# Improving and Assessing Anomaly Detectors for Large-Scale Settings

## Abstract

Detecting out-of-distribution examples is important for safety-critical machine learning applications such as detecting novel biological phenomena and self-driving cars. However, existing research mainly focuses on simple small-scale settings. To set the stage for more realistic out-of-distribution detection, we depart from small-scale settings and explore large-scale multiclass and multi-label settings with high-resolution images and thousands of classes. To make future work in real-world settings possible, we create new benchmarks for three large-scale settings. To test ImageNet multiclass anomaly detectors, we introduce a new dataset of anomalous species. We leverage ImageNet-21K to evaluate PASCAL VOC and COCO multilabel anomaly detectors. Third, we introduce a new benchmark for anomaly segmentation by introducing a segmentation benchmark with road anomalies. We conduct extensive experiments in these more realistic settings for out-of-distribution detection and find that a surprisingly simple detector based on the maximum logit outperforms prior methods in all the large-scale multi-class, multi-label, and segmentation tasks, establishing a simple new baseline for future work.

## 1 Introduction

Out-of-distribution (OOD) detection is a valuable tool for developing safe and reliable machine learning (ML) systems. Detecting anomalous inputs allows systems to initiate a conservative fallback policy or defer to human judgment. As an important component of ML Safety (Hendrycks et al., 2021), OOD detection is important for safety-critical applications such as self-driving cars and detecting novel microorganisms. Accordingly, research on out-of-distribution detection has a rich history spanning several decades (Schölkopf et al., 1999; Breunig et al., 2000; Emmott et al., 2015). Recent work leverages deep neural representations for out-of-distribution detection in complex domains, such as image data (Hendrycks & Gimpel, 2017; Lee et al., 2018a; Mohseni et al., 2020; Hendrycks et al., 2019b). However, these works still primarily use small-scale datasets with low-resolution images and few classes. As the community moves towards more realistic, large-scale settings, strong baselines and high-quality benchmarks are imperative for future progress.

Large-scale datasets such as ImageNet (Deng et al., 2009) and Places365 (Zhou et al., 2017) present unique challenges not seen in small-scale settings, such as a plethora of fine-grained object classes. We demonstrate that the maximum softmax probability (MSP) detector, a state-of-the-art method for small-scale problems, does not scale well to these challenging conditions. Through extensive experiments, we identify a detector based on the maximum logit (MaxLogit) that greatly outperforms the MSP and other strong baselines in large-scale multi-class anomaly segmentation. To facilitate further research in this setting, we also collect a new out-of-distribution test dataset suitable for models trained on highly diverse datasets. Shown in Figure 2, our Species dataset contains diverse, anomalous species that do not overlap ImageNet-21K which has approximately twenty two thousand classes. Species avoids data leakage and enables a stricter evaluation methodology for ImageNet-21K models. Using Species to conduct more controlled experiments without train-test overlap, we find that contrary to prior claims (Fort et al., 2021; Koner et al., 2021), Vision Transformers (Dosovitskiy et al., 2021a) pre-trained on ImageNet-21K are not substantially better at out-of-distribution detection.

Moreover, in the common real-world case of multi-label data, the MSP detector cannot naturally be applied in the first place, as it requires softmax probabilities. To enable research into the multi-label setting for anomaly detection, we contribute a multi-label experimental setup and explore various

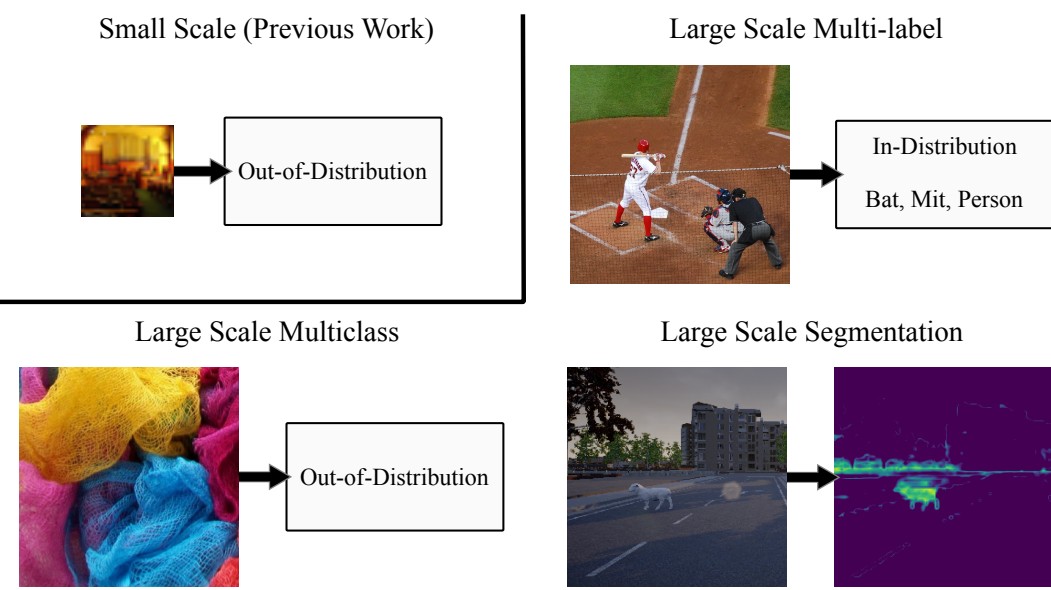

Figure 1: We scale up out-of-distribution detection to large-scale multi-class datasets with thousands of classes, multi-label datasets with complex scenes, and anomaly segmentation in driving environments. We introduce new benchmarks for all three settings. In all of these settings, we find that an OOD detector based on the maximum logit outperforms previous methods, establishing a strong and versatile baseline for future work on large-scale OOD detection. The bottom-right shows a scene from our new anomaly segmentation benchmark and the predicted anomaly using a state-of-the-art detector.

methods on large-scale multi-label datasets. We find that the MaxLogit detector from our investigation into the large-scale multi-class setting generalizes well to multi-label data and again outperforms all other baselines.

In addition to focusing on small-scale datasets, most existing benchmarks for anomaly detection treat entire images as anomalies. In practice, an image could be anomalous in localized regions while being in-distribution elsewhere. Knowing which regions of an image are anomalous could allow for safer handling of unfamiliar objects in the case of self-driving cars. Creating a benchmark for this task is difficult, though, as simply cutting and pasting anomalous objects into images introduces various unnatural giveaway cues such as edge effects, mismatched orientation, and lighting, all of which trivialize the task of anomaly segmentation (Blum et al., 2019).

To overcome these issues, we utilize a simulated driving environment to create the novel StreetHazards dataset for anomaly segmentation. Using the Unreal Engine and the open-source CARLA simulation environment (Dosovitskiy et al., 2017), we insert a diverse array of foreign objects into driving scenes and re-render the scenes with these novel objects. This enables integration of the foreign objects into their surrounding context with correct lighting and orientation, sidestepping giveaway cues.

To complement the StreetHazards dataset, we convert the BDD100K semantic segmentation dataset (Yu et al., 2018) into an anomaly segmentation dataset, which we call BDD-Anomaly. By leveraging the large scale of BDD100K, we reserve infrequent object classes to be anomalies. We combine this dataset with StreetHazards to form the Combined Anomalous Object Segmentation (CAOS) benchmark. The CAOS benchmark improves over previous evaluations for anomaly segmentation in driving scenes by evaluating detectors on realistic and diverse anomalies. We evaluate several baselines on the CAOS benchmark and discuss problems with porting existing approaches from earlier formulations of out-of-distribution detection.

Despite its simplicity, we find that the MaxLogit detector outperforms all baselines on Species, our multi-class benchmark, and CAOS. In each of these three settings, we discuss why MaxLogit provides superior performance, and we show that these gains are hidden if one looks at small-scale problems alone. The code for our experiments and the Species and CAOS datasets are available at [anonymized]. Our new baseline combined with Species and CAOS benchmarks pave the way for future research on large-scale out-of-distribution detection.

Anomalous Species Dataset

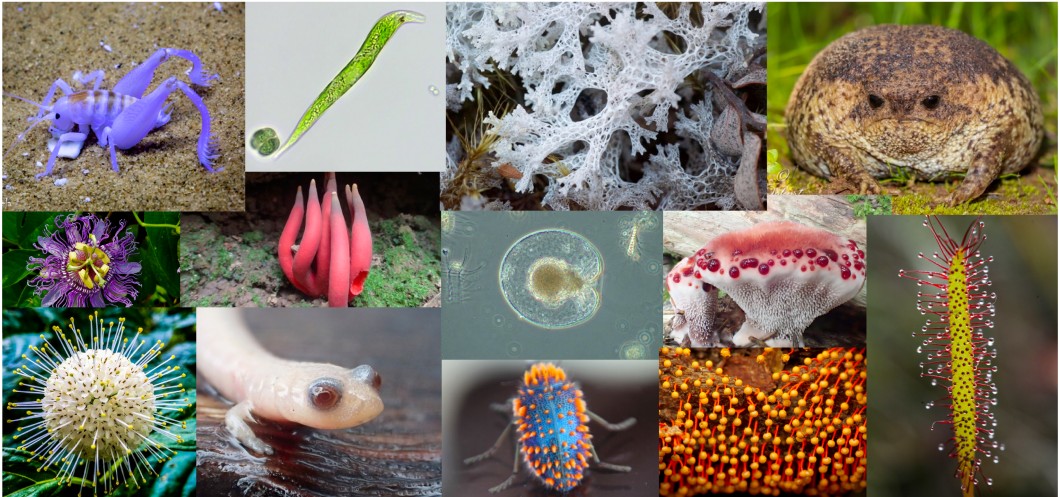

Figure 2: The Species out-of-distribution dataset is designed for large-scale anomaly detectors pretrained on datasets as diverse as ImageNet-21K. When models are pretrained on ImageNet-21K, many previous OOD detection datasets may overlap with the pretraining set, resulting in erroneous evaluations. To rectify this, Species is comprised of hundreds of anomalous species that are disjoint from ImageNet-21K classes and enables the evaluation of cutting-edge models.

## 2   RELATED WORK

**Multi-Class Out-of-Distribution Detection.**   A recent line of work leverages deep neural representations from multi-class classifiers to perform out-of-distribution (OOD) detection on high-dimensional data, including images, text, and speech data. Hendrycks & Gimpel (2017) formulate the task and propose the simple baseline of using the maximum softmax probability of the classifier on an input to gauge whether the input is out-of-distribution. In particular, they formulate the task as distinguishing between examples from an in-distribution dataset and various OOD datasets. Importantly, entire images are treated as out-of-distribution.

Continuing this line of work, Lee et al. (2018a) propose to improve the neural representation of the classifier to better separate OOD examples. They use generative adversarial networks to produce near-distribution examples and induce uniform posteriors on these synthetic OOD examples. Hendrycks et al. (2019b) observe that outliers are often easy to obtain in large quantity from diverse, realistic datasets and demonstrate that OOD detectors trained on these outliers generalize to unseen classes of anomalies. Other work investigates improving the anomaly detectors themselves given a fixed classifier (DeVries & Taylor, 2018; Liang et al., 2018). However, as Hendrycks et al. (2019b) observe, many of these works tune hyperparameters on a particular type of anomaly that is also seen at test time, so their evaluation setting is more lenient. In this paper, all anomalies seen at test time come from entirely unseen categories and are not tuned on in any way. Hence, we do not compare to techniques such as ODIN (Liang et al., 2018). Additionally, in a point of departure from prior work, we focus primarily on large-scale images and datasets with many classes.

Recent work has suggested that stronger representations from Vision Transformers pre-trained on ImageNet-21K can make out-of-distribution detection trivial (Fort et al., 2021; Koner et al., 2021). They evaluate models on detecting CIFAR-10 when fine-tuned on CIFAR-100 or vice versa, using models pretrained on ImageNet-21K. However, over 1,000 classes in ImageNet-21K overlap with CIFAR-10, so it is still unclear how Vision Transformers perform at detecting entirely unseen OOD categories. We create a new OOD test dataset of anomalous species to investigate how well Vision Transformers perform in controlled OOD detection settings without data leakage and overlap. We find that Vision Transformers pre-trained on ImageNet-21K are far from solving OOD detection in large-scale settings.

| $\mathcal{D}_{\text{in}}$ | FPR95 ↓ | | | AUROC ↑ | | | AUPR ↑ | | |
|---|---|---|---|---|---|---|---|---|---|
| | MSP | DeVries | MaxLogit | MSP | DeVries | MaxLogit | MSP | DeVries | MaxLogit |
| ImageNet | 44.2 | 46.0 | **35.8** | 84.6 | 76.9 | **87.2** | 38.2 | 30.5 | **45.8** |
| Places365 | 52.6 | 85.8 | **36.6** | 76.0 | 31.1 | **85.8** | 8.2 | 2.0 | **19.2** |

Table 1: Multi-class out-of-distribution detection results using the maximum softmax probability (MSP) baseline (Hendrycks & Gimpel, 2017), the confidence branch detector of DeVries & Taylor (2018), and our maximum logit baseline. All values are percentages and average across five out-of-distribution test datasets. Full results on individual OOD test datasets are in the Appendix.

**Anomaly Segmentation.** Several prior works explore segmenting anomalous image regions. One line of work uses the WildDash dataset (Zendel et al., 2018), which contains numerous annotated driving scenes in conditions such as snow, fog, and rain. The WildDash test set contains fifteen "negative images" from different domains for which the goal is to mark the entire image as out-of-distribution. Thus, while the task is segmentation, the anomalies do not exist as objects within an otherwise in-distribution scene. This setting is similar to that explored by Hendrycks & Gimpel (2017), in which whole images from other datasets serve as out-of-distribution examples.

To approach anomaly segmentation on WildDash, Krešo et al. (2018) train on multiple semantic segmentation domains and treat regions of images from the WildDash driving dataset as out-of-distribution if they are segmented as regions from different domains, i.e. indoor classes. Bevandić et al. (2018) use ILSVRC 2012 images and train their network to segment the entirety of these images as out-of-distribution.

In medical anomaly segmentation and product fault detection, anomalies are regions of otherwise in-distribution images. Baur et al. (2019) segment anomalous regions in brain MRIs using pixel-wise reconstruction loss. Similarly, Haselmann et al. (2018) perform product fault detection using pixel-wise reconstruction loss and introduce an expansive dataset for segmentation of product faults. In these relatively simple domains, reconstruction-based approaches work well. In contrast to medical anomaly segmentation and fault detection, we consider complex images from street scenes. These images have high variability in scene layout and lighting, and hence are less amenable to reconstruction-based techniques.

The two works closest to our own are the Lost and Found (Pinggera et al., 2016) and Fishyscapes (Blum et al., 2019) datasets. The Lost and Found dataset consists of real images in a driving environment with small road hazards. The images were collected to mirror the Cityscapes dataset (Cordts et al., 2016) but are only collected from one city and so have less diversity. The dataset contains 35 unique anomalous objects, and methods are allowed to train on many of these. For Lost and Found, only nine unique objects are truly unseen at test time. Crucially, this is a different evaluation setting from our own, where anomalous objects are not revealed at training time, so their dataset is not directly comparable. Nevertheless, the BDD-Anomaly dataset fills several gaps in Lost and Found. First, the images are more diverse, because they are sourced from a more recent and comprehensive semantic segmentation dataset. Second, the anomalies are not restricted to small, sparse road hazards. Concretely, anomalous regions in Lost and Found take up 0.11% of the image on average, whereas anomalous regions in the BDD-Anomaly dataset are larger and fill 0.83% of the image on average. Finally, although the BDD-Anomaly dataset treats three categories as anomalous, compared to Lost and Found it has far more unique anomalous objects.

The Fishyscapes benchmark for anomaly segmentation consists of cut-and-paste anomalies from out-of-distribution domains. This is problematic, because the anomalies stand out as clearly unnatural in context. For instance, the orientation of anomalous objects is unnatural, and the lighting of the cut-and-paste patch differs from the lighting in the original image, providing an unnatural cue to anomaly detectors that would not exist for real anomalies. Figure 7 shows an example of these inconsistencies. Techniques for detecting image manipulation (Zhou et al., 2018; Johnson & Farid, 2005) are competent at detecting artificial image elements of this kind. Our StreetHazards dataset overcomes these issues by leveraging a simulated driving environment to naturally insert anomalous *3D models* into a scene rather than overlaying 2D images. These anomalies are integrated into the scene with proper lighting and orientation, mimicking real-world anomalies and making them significantly more difficult to detect.

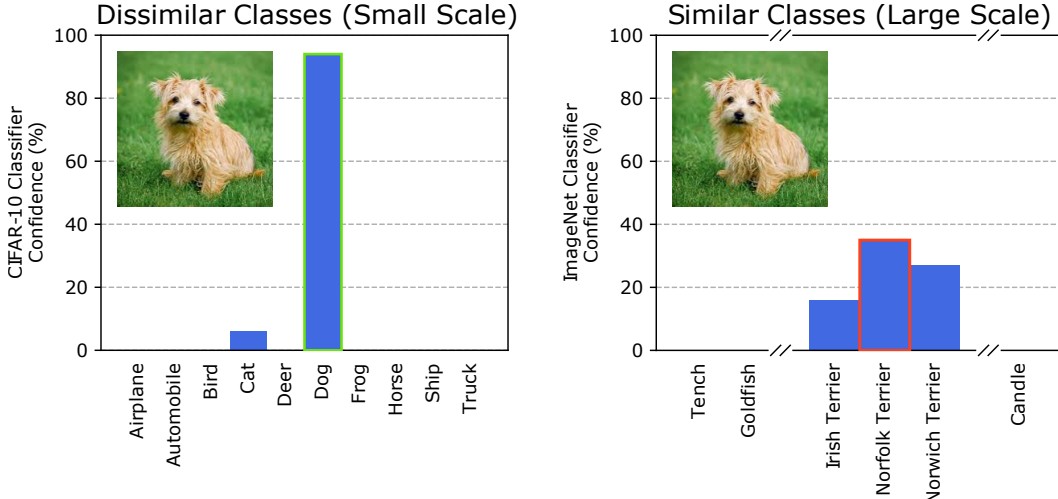

Figure 3: Small-scale datasets such as CIFAR-10 have relatively disjoint classes, but larger-scale datasets including ImageNet-1K have several classes with high visual similarity to other classes. This implies that large-scale classifiers disperse probability mass among several classes. If the prediction confidence is used for out-of-distribution detection, then images which have similarities to other classes will often wrongly be deemed out-of-distribution due to low and dispersed confidence. This motivates our MaxLogit out-of-distribution detector.

## 3 MULTI-CLASS PREDICTION FOR OOD DETECTION

**Problem with existing baselines.** Existing baselines for anomaly detection can work well in small-scale settings. However, in more realistic settings image classification networks are often tasked with distinguishing hundreds or thousands of classes, possibly with subtle differences. This is problematic for the maximum softmax probability (MSP) baseline (Hendrycks & Gimpel, 2017), which uses the negative maximum softmax probability as the anomaly score, or $-\max_k \exp f(x)_k / \sum_i \exp f(x)_i = -\max_k \hat{p}(y = k \mid x)$, where $f(x)$ is the unnormalized logits of classifier $f$ on input $x$. Classifiers tend to have higher confidence on in-distribution examples than out-of-distribution examples, enabling OOD detection. Assuming single-model evaluation and no access to other anomalies or test-time adaptation, the MSP attains state-of-the-art anomaly detection performance in small-scale settings. However, we show that the MSP is problematic for realistic in-distribution datasets with many classes, such as ImageNet and Places365 (Zhou et al., 2017). Probability mass can be dispersed among visually similar classes, as shown in Figure 3. Consequently, a classifier may produce a low confidence prediction for an in-distribution image, not because the image is unfamiliar, but because the object's exact class is difficult to determine. To circumvent this problem, we propose using the negative of the maximum unnormalized logit for an anomaly score $-\max_k f(x)_k$, which we call MaxLogit. Since the logits are unnormalized, they are not affected by the number of classes and can serve as a better baseline for large-scale out-of-distribution detection.

**The Species Out-Of-Distribution Dataset.** To enable controlled experiments and high-quality evaluations of anomaly detectors in large-scale settings, we create the Species dataset, a new out-of-distribution test dataset that has no overlapping classes with ImageNet-21K. The Species dataset is comprised of images scraped from the iNaturalist website and contains hundreds of anomalous species grouped into seven high-level categories: Plants, Microorganisms, Amphibians, Protozoa, Fungi, Arachnids, and Insects. Example images from the Species dataset are in Figure 2.

**Setup.** To evaluate the MSP baseline out-of-distribution detector and the MaxLogit detector, we use ImageNet-21K as the in-distribution dataset $\mathcal{D}_{\text{in}}$. To obtain representations for anomaly detection, we use models trained on ImageNet-21K-P, a cleaned version of ImageNet-21K with a train/val split (Ridnik et al., 2021a). We evaluate a TResNet-M, ViT-B-16, and Mixer-B-16 (Ridnik et al., 2021b; Dosovitskiy et al., 2021b; Tolstikhin et al., 2021), and the validation split is used for obtaining in-distribution scores. For out-of-distribution test datasets $\mathcal{D}_{\text{out}}$, we use categories from the Species dataset, all of which are unseen during training. Results for these experiments are in Table 2. We

| $\mathcal{D}_{\text{in}}$ | $\mathcal{D}_{\text{out}}^{\text{test}}$ | ResNet | | ViT | | MLP Mixer | |
|---|---|---|---|---|---|---|---|
| | | MSP | MaxLogit | MSP | MaxLogit | MSP | MaxLogit |
| ImageNet-21K-P | Plants | 80.3 | 87.8 | 78.2 | 84.8 | 80.3 | 85.0 |
| | Microorganisms | 77.4 | 83.4 | 71.1 | 82.4 | 74.4 | 86.0 |
| | Amphibians | 41.8 | 48.6 | 41.9 | 48.8 | 44.4 | 51.7 |
| | Protozoa | 70.7 | 80.4 | 69.3 | 80.9 | 68.0 | 77.7 |
| | Fungi | 66.4 | 77.4 | 64.7 | 76.1 | 64.1 | 76.9 |
| | Arachnids | 46.9 | 56.7 | 46.6 | 56.8 | 48.9 | 58.8 |
| | Insects | 47.6 | 56.4 | 48.0 | 54.6 | 48.6 | 53.8 |
| | Mean | 61.6 | **70.1** | 60.0 | **69.2** | 61.2 | **70.0** |

Table 2: Results on Species. Models and the processed version of ImageNet-21K (ImageNet-21K-P) are from Ridnik et al. (2021a). All values are percent AUROC. Species enables evaluating anomaly detectors trained on ImageNet-21K and evades class overlap issues present in prior work. Using Species to conduct more controlled experiments without class overlap issues, we find that contrary to recent claims (Fort et al., 2021), simply scaling up Vision Transformers does not make OOD detection trivial.

also use ImageNet-1K and Places365 as in-distribution datasets $\mathcal{D}_{\text{in}}$, for which we use pretrained ResNet-50 models and use several out-of-distribution test datasets $\mathcal{D}_{\text{out}}$. Full results with ImageNet and Places365 as in-distribution are in the Appendix.

**Metrics.** To evaluate out-of-distribution detectors in large-scale settings, we use three standard metrics of detection performance: area under the ROC curve (AUROC), false positive rate at 95% recall (FPR95), and area under the precision-recall curve (AUPR). The AUROC and AUPR are important metrics, because they give a holistic measure of performance when the cutoff for detecting anomalies is not a priori obvious or when we want to represent the performance of a detection method across several different cutoffs.

The AUROC can be thought of as the probability that an anomalous example is given a higher score than an ordinary example. Thus, a higher score is better, and an uninformative detector has a AUROC of 50%. AUPR provides a metric more attuned to class imbalances, which is relevant in anomaly and failure detection, when the number of anomalies or failures may be relatively small. Last, the FPR95 metric consists of measuring the false positive rate at 95%. Since these measures are correlated, we occasionally solely present the AUROC for brevity and to preserve space.

**Results.** Results on Species are shown in Table 2. Results with ImageNet-1K and Places365 as in-distribution datasets are in Table 1. We find that the proposed MaxLogit method outperforms the maximum softmax probability baseline on all out-of-distribution test datasets $\mathcal{D}_{\text{out}}$. This holds true for all three models trained on ImageNet-21K. The MSP baseline is not much better than random and is has similar performance for all three model classes. This suggests that contrary to recent claims, (Fort et al., 2021) simply scaling up Vision Transformers does not make OOD detection trivial.

## 4 MULTI-LABEL PREDICTION FOR OOD DETECTION

Current work on out-of-distribution detection primarily considers multi-class or unsupervised settings. Yet as classifiers become more useful in realistic settings, the multi-label formulation becomes increasingly natural. To investigate out-of-distribution detection in multi-label settings, we provide a baseline and evaluation setup.

**Setup.** For multi-label classification we use PASCAL VOC (Everingham et al., 2009) and MS-COCO (Lin et al., 2014) as in-distribution data. To evaluate anomaly detectors for these in-distribution datasets, we use 20 out-of-distribution classes from ImageNet-21K. These classes have no overlap with ImageNet-1K, PASCAL VOC, or MS-COCO. The 20 classes are chosen not to overlap with ImageNet-1K since the multi-label classifiers models are pre-trained on ImageNet-1K. We list the class WordNet IDs in the Appendix.

**Methods.** For our experiments, we use a ResNet-101 backbone architecture pre-trained on ImageNet-1K. We replace the final layer with 2 fully connected layers and apply the logistic sigmoid function for multi-label prediction. During training we freeze the batch normalization parameters due to an insufficient number of images for proper mean and variance estimation. We train each model for 50 epochs using the Adam optimizer (Kingma & Ba, 2014) with hyperparameter values $10^{-4}$ and

|  |  |  | iForest | LOF | Dropout | LogitAvg | MSP | MaxLogit |
|---|---|---|---|---|---|---|---|---|
| | FPR95 | ↓ | 98.6 | 84.0 | 97.2 | 98.2 | 82.3 | **35.6** |
| PASCAL VOC | AUROC | ↑ | 46.3 | 68.4 | 49.2 | 47.9 | 74.2 | **90.9** |
| | AUPR | ↑ | 37.1 | 58.4 | 45.3 | 41.3 | 65.5 | **81.2** |
| | FPR95 | ↓ | 95.6 | 78.4 | 93.3 | 94.5 | 81.8 | **40.4** |
| COCO | AUROC | ↑ | 41.4 | 70.2 | 58.0 | 55.5 | 70.7 | **90.3** |
| | AUPR | ↑ | 63.7 | 82.0 | 76.3 | 74.0 | 82.9 | **94.0** |

Table 3: Multi-label out-of-distribution detection comparison of the Isolation Forest (iForest), Local Outlier Factor (LOF), Dropout, logit average, maximum softmax probability, and maximum logit anomaly detectors on PASCAL VOC and MS-COCO. The same network architecture is used for all three detectors. All results shown are percentages.

$10^{-5}$ for $\beta_1$ and $\beta_2$ respectively. For data augmentation we use standard resizing, random crops, and random flips to obtain images of size $256 \times 256 \times 3$. As a result of this training procedure, the mAP of the ResNet-101 on PASCAL VOC is 89.11% and 72.0% for MS-COCO.

As there has been little work on out-of-distribution detection in multilabel settings, we include comparisons to classic anomaly detectors for general settings. Isolation Forest, denoted by iForest, works by randomly partitioning the space into half spaces to form a decision tree. The score is determined by how close a point is to the root of the tree. The local outlier factor (LOF) (Breunig et al., 2000) computes a local density ratio between every element and its neighbors. We set the number of neighbors as 20. iForest and LOF are both computed on features from the penultimate layer of the networks. MSP denotes a natural extension of the maximum softmax probability detector in the multi-label setting, obtained by taking the sigmoid of each output score $f(x)_i$ and computing $-\max_i \sigma(f(x)_i)$. Alternatively, one can average the logit values, denoted by LogitAvg. These serve as our baseline detectors for multi-label OOD detection. We compare these baselines to the MaxLogit detector that we introduce in Section 3. As in the multi-class case, the MaxLogit anomaly score for multi-label classification is $-\max_i f(x)_i$.

**Results.** Results are shown in Table 3. We find that MaxLogit obtains the highest performance in all cases. MaxLogit bears similarity to the MSP baseline (Hendrycks & Gimpel, 2017) but is naturally applicable to multi-label problems. These results establish the MaxLogit as an effective and natural baseline for large-scale multi-label problems. Further, the evaluation setup enables future work in out-of-distribution detection with multi-label datasets.

## 5 THE CAOS BENCHMARK

The Combined Anomalous Object Segmentation (CAOS) benchmark is comprised of two complementary datasets for evaluating anomaly segmentation systems on diverse, realistic anomalies. First is the StreetHazards dataset, which leverages simulation to provide a large variety of anomalous objects realistically inserted into driving scenes. Second is the BDD-Anomaly dataset, which consists of real images taken from the BDD100K dataset (Yu et al., 2018). StreetHazards contains a highly diverse array of anomalies; BDD-Anomaly contains anomalies in real-world images. Together, these datasets allow researchers to judge techniques on their ability to segment diverse anomalies as well as anomalies in real images. All images have $720 \times 1280$ resolution.

**The StreetHazards Dataset.** StreetHazards is an anomaly segmentation dataset that leverages simulation to provide diverse, realistically-inserted anomalous objects. To create the StreetHazards dataset, we use the Unreal Engine along with the CARLA simulation environment (Dosovitskiy et al., 2017). From several months of development and testing including customization of the Unreal Engine and CARLA, we can insert foreign entities into a scene while having them be properly integrated. Unlike previous work, this avoids the issues of inconsistent chromatic aberration, inconsistent lighting, edge effects, and other simple cues that an object is anomalous. Additionally, using a simulated environment allows us to dynamically insert diverse anomalous objects in any location and have them render properly with changes to lighting and weather including time of day, cloudy skies, and rain.

We use 3 towns from CARLA for training, from which we collect RGB images and their respective semantic segmentation maps to serve as training data for semantic segmentation models. We generate a validation set from the fourth town. Finally, we reserve the fifth and sixth town as our test set. We insert anomalies taken from the Digimation Model Bank Library and semantic ShapeNet

Examples and Predictions for Our StreetHazards Dataset

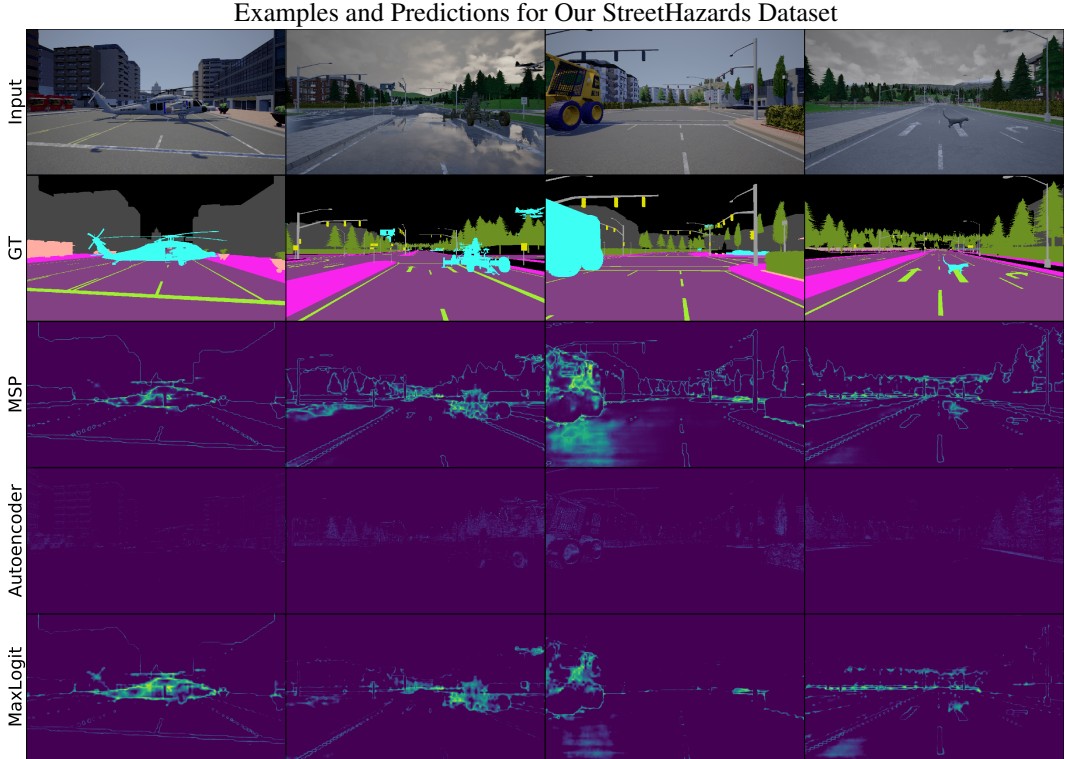

Figure 4: A sample of anomalous scenes from the CAOS benchmark with model predictions and anomaly scores. The anomaly scores are thresholded to the top 10% of values for visualization. GT is ground truth, the autoencoder model is based on the spatial autoencoder used in Baur et al. (2019), MSP is the maximum softmax probability baseline (Hendrycks & Gimpel, 2017), and MaxLogit is the method we propose as a new baseline for large-scale settings. Compared to baselines, the MaxLogit detector places lower scores on in-distribution image regions, including object outlines, while also doing a better job of highlighting anomalous objects.

(ShapeNetSem) (Savva et al., 2015) into the test set in order to evaluate methods for out-of-distribution detection. In total, we use 250 unique anomaly models of diverse types. There are 12 classes used for training: background, road, street lines, traffic signs, sidewalk, pedestrian, vehicle, building, wall, pole, fence, and vegetation. The thirteenth class is the anomaly class that is only used at test time. We collect 5,125 image and semantic segmentation ground truth pairs for training, 1,031 pairs without anomalies for validation, and 1,500 test pairs with anomalies.

**The BDD-Anomaly Dataset.** BDD-Anomaly is an anomaly segmentation dataset with real images in diverse conditions. We source BDD-Anomaly from BDD100K (Yu et al., 2018), a large-scale semantic segmentation dataset with diverse driving conditions. The original data consists in 7,000 images for training and 1,000 for validation. There are 18 original classes. We choose *motorcycle*, *train*, and *bicycle* as the anomalous object classes and remove all images with these objects from the training and validation sets. This yields 6,280 training pairs, 910 validation pairs without anomalies, and 810 testing pairs with anomalous objects.

## 5.1 EXPERIMENTS

**Evaluation.** In anomaly segmentation experiments, each pixel is treated as a prediction, resulting in many predictions to evaluate. To fit these in memory, we compute the metrics on each image and average over the images to obtain final values.

**Methods.** Our first baseline is pixel-wise Maximum Softmax Probability (MSP). Introduced by Hendrycks & Gimpel (2017) for multi-class out-of-distribution detection, we directly port this baseline to anomaly segmentation. Alternatively, the background class might serve as an anomaly detector, because it contains everything not in the other classes. To test this hypothesis, "Background" uses the posterior probability of the background class as the anomaly score. The Dropout method

|  | | MSP | Branch | Background | Dropout | AE | MaxLogit |
|---|---|---|---|---|---|---|---|
| StreetHazards | FPR95 ↓ | 33.7 | 68.4 | 69.0 | 79.4 | 91.7 | **26.5** |
|  | AUROC ↑ | 87.7 | 65.7 | 58.6 | 69.9 | 66.1 | **89.3** |
|  | AUPR ↑ | 6.6 | 1.5 | 4.5 | 7.5 | 2.2 | **10.6** |
| BDD-Anomaly | FPR95 ↓ | 24.5 | 25.6 | 40.1 | 16.6 | 74.1 | **14.0** |
|  | AUROC ↑ | 87.7 | 85.6 | 69.7 | 90.8 | 64.0 | **92.6** |
|  | AUPR ↑ | 3.7 | 3.9 | 1.1 | 4.3 | 0.7 | **5.4** |

Table 4: Results on the CAOS benchmark. AUPR is low across the board due to the large class imbalance, but all methods perform substantially better than chance. MaxLogit obtains the best performance. All results are percentages.

leverages MC Dropout (Gal & Ghahramani, 2016) to obtain an epistemic uncertainty estimate. Following Kendall et al. (2015), we compute the pixel-wise posterior variance over multiple dropout masks and average across all classes, which serves as the anomaly score. We also experiment with an autoencoder baseline similar to Baur et al. (2019); Haselmann et al. (2018) where pixel-wise reconstruction loss is used as the anomaly score. This method is called AE. The "Branch" method is a direct port of the confidence branch detector from DeVries & Taylor (2018) to pixel-wise prediction. Finally, we use the MaxLogit method described in earlier sections independently on each pixel.

For all of the baselines except the autoencoder, we train a PSPNet (Zhao et al., 2017) decoder with a ResNet-101 encoder (He et al., 2015) for 20 epochs. We train both the encoder and decoder using SGD with momentum of 0.9, a learning rate of $2 \times 10^{-2}$, and learning rate decay of $10^{-4}$. For AE, we use a 4-layer U-Net (Ronneberger et al., 2015) with a spatial latent code as in Baur et al. (2019). The U-Net also uses batch norm and is trained for 10 epochs. Results are in Table 4.

**Results and Analysis.** MaxLogit outperforms all other methods across the board by a substantial margin. The intuitive baseline of using the posterior for the background class to detect anomalies performs poorly, which suggests that the background class may not align with rare visual features. Even though reconstruction-based scores succeed in product fault segmentation, we find that the AE method performs poorly on the CAOS benchmark, which may be due to the more complex domain. AUPR for

| Method | MSP | MaxLogit |
|---|---|---|
| FS Lost and Found | 87.0% | 92.0% |
| Road Anomaly | 73.8% | 78.0% |

Figure 5: Auxiliary analysis of the MSP and the MaxLogit AUROCs using prior less comprehensive anomaly segmentation datasets.

all methods is low, indicating that the large class imbalance presents a serious challenge. However, the substantial improvements with the MaxLogit method suggest that progress on this task is possible and there is much room for improvement. A comparison with other datasets is in Figure 5 (Pinggera et al., 2016; Blum et al., 2019; Jung et al., 2021).

In Figure 4, we see that both MaxLogit and MSP have many false positives, as they assign high anomaly scores to semantic boundaries, a problem also observed in the recent works of (Blum et al., 2019; Angus, 2019). However, the problem is less severe with MaxLogit. A potential explanation for this is that even when the prediction confidence dips at semantic boundaries, the maximum logit can remain the same in a 'hand-off' procedure between the classes. Thus, MaxLogit provides a natural mechanism to combat semantic boundary artifacts that could be further explored in future work.

## 6 CONCLUSION

We scaled out-of-distribution detection to settings with thousands of classes and high-resolution images. We identified an issue faced by existing baselines when scaling to these settings and proposed the maximum logit detector as a natural solution. We introduced the Species dataset to enable more controlled experiments without class overlap and also investigated using multi-label classifiers for OOD detection, establishing an experimental setup for this previously unexplored setting. Finally, we introduced the CAOS benchmark for anomaly segmentation, consisting of diverse, naturally-integrated anomalous objects in driving scenes. Baseline methods on the CAOS benchmark substantially improve on random guessing but are still lacking, indicating potential for future work. Interestingly, the MaxLogit detector also provides consistent and significant gains in the multi-label and anomaly segmentation settings, thereby establishing it as a new baseline in place of the maximum softmax prob-

ability baseline on large-scale OOD detection problems. In all, we we hope that our contributions will enable further research on out-of-distribution detection for real-world safety-critical environments.

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

# A  APPENDIX

| $\mathcal{D}_{\text{in}}$ | $\mathcal{D}_{\text{out}}^{\text{test}}$ | FPR95 ↓ | | | | AUROC ↑ | | | | AUPR ↑ | | | |
|---|---|---|---|---|---|---|---|---|---|---|---|---|---|
| | | B | M | D | K | B | M | D | K | B | M | D | K |
| ImageNet | Gaussian | 2 | 0 | 5 | 4 | 100 | 100 | 97 | 98 | 93 | 98 | 55 | 79 |
| | Rademacher | 21 | 4 | 4 | 15 | 89 | 98 | 98 | 93 | 29 | 70 | 62 | 54 |
| | Blobs | 26 | 32 | 72 | 8 | 80 | 79 | 37 | 99 | 25 | 17 | 7 | 93 |
| | Textures | 68 | 56 | 74 | 59 | 80 | 87 | 76 | 85 | 25 | 36 | 16 | 48 |
| | LSUN | 66 | 63 | 59 | 60 | 75 | 77 | 76 | 79 | 21 | 22 | 19 | 38 |
| | Places365 | 64 | 59 | 63 | 72 | 79 | 83 | 79 | 79 | 27 | 32 | 24 | 46 |
| | Mean | 41.3 | 35.8 | 46 | 36.1 | 85.2 | 87.2 | 76.9 | 88.7 | 37 | 45.8 | 30.5 | 59.7 |
| Places365 | Gaussian | 10 | 6 | 71 | 12 | 93 | 96 | 35 | 93 | 16 | 24 | 2 | 16 |
| | Rademacher | 20 | 10 | 91 | 1 | 89 | 93 | 10 | 100 | 11 | 15.9 | 1.6 | 88 |
| | Blobs | 59 | 6 | 88 | 27 | 72 | 98 | 15 | 93 | 5 | 41 | 2 | 31 |
| | Textures | 86 | 72 | 87 | 74 | 65 | 79 | 43 | 79 | 4 | 11 | 1 | 12 |
| | Places69 | 88 | 89 | 92 | 91 | 61 | 64 | 52 | 65 | 5 | 6 | 3 | 6 |
| | Mean | 53 | 36.6 | 85.8 | 40.9 | 76 | 85.8 | 31.1 | 85.8 | 8 | 19.2 | 2 | 30.5 |

Table 5: B is for the maximum softmax probability baseline, M is for maximum logit, D is for the method in DeVries & Taylor (2018), and K is our own KL method described below. Both M and K are ours. Results are on ImageNet and Places365. All values are percentages and are rounded so that 99.95 rounds to 100.

# B  FULL MULTICLASS OOD DETECTION RESULTS

**Datasets.**  To evaluate the MSP baseline out-of-distribution detector and the MaxLogit detector, we use the ImageNet-1K object recognition dataset and Places365 scene recognition dataset as in-distribution datasets $\mathcal{D}_{\text{in}}$. We use several out-of-distribution test datasets $\mathcal{D}_{\text{out}}$, all of which are unseen during training. The first out-of-distribution dataset is *Gaussian* noise, where each example's pixels are i.i.d. sampled from $\mathcal{N}(0, 0.5)$ and clipped to be contained within $[-1, 1]$. Another type of test-time noise is *Rademacher* noise, in which each pixel is i.i.d. sampled from $2 \cdot \text{Bernoulli}(0.5) - 1$, i.e. each pixel is 1 or $-1$ with equal probability. *Blob* examples are more structured than noise; they are algorithmically generated blob images. Meanwhile, *Textures* is a dataset consisting in images of describable textures (Cimpoi et al., 2014). When evaluating the ImageNet-1K detector, we use *LSUN* images, a scene recognition dataset (Yu et al., 2015). Our final $\mathcal{D}_{\text{out}}$ is *Places69*, a scene classification dataset that does not share classes with Places365. In all, we evaluate against out-of-distribution examples spanning synthetic and realistic images.

**KL Matching Method.**  To verify our intuitions that led us to develop the MaxLogit detector, we developed a less convenient but similarly powerful technique applicable for the multiclass setting. Recall that some classes tend to be predicted with low confidence and others high confidence. The shape of predicted posterior distributions is often class dependent.

We capture the typical shape of each class's posterior distribution and form posterior distribution templates for each class. During test time, the network's softmax posterior distribution is compared to these templates and an anomaly score is generated. More concretely, we compute $k$ different distributions $d_k$, one for each class. We write $d_k = \mathbb{E}_{x' \sim \mathcal{X}_{\text{val}}}[p(y|x')]$ where $k = \text{argmax}_k\, p(y = k \mid x')$. Then for a new test input $x$, we calculate the anomaly score $\min_k \text{KL}[p(y \mid x) \parallel d_k]$ rather than the MSP baseline $-\max_k p(y = k \mid x)$. Note that we utilize the validation dataset, but our KL matching method does not require the validation dataset's labels. That said, our KL matching method is less convenient than our MaxLogit technique, and the two perform similarly. Since this technique requires more data than MaxLogit, we opt to simply use the MaxLogit in the main paper.

**Results.**  Observe that the proposed MaxLogit method outperforms the maximum softmax probability baseline for all three metrics on both ImageNet and Places365. These results were computed using a ResNet-50 trained on either ImageNet-1K or Places365. In the case of Places365, the AUROC improvement is over 10%. We note that the utility of the maximum logit could not be appreciated as easily in previous work's small-scale settings. For example, using the small-scale CIFAR-10 setup of

Hendrycks et al. Hendrycks et al. (2019a), the MSP attains an average AUROC of 90.08% while the maximum logit attains 90.22%, a minor 0.14% difference. However, in a large-scale setting, the difference can be over 10% on individual $\mathcal{D}_{\text{out}}$ datasets. We are not claiming that utilizing the maximum logit is a mathematically innovative formulation, only that it serves as a consistently powerful baseline for large-scale settings that went unappreciated in small-scale settings. In consequence, we suggest using the maximum logit as a new baseline for large-scale multi-class out-of-distribution detection.

**Overview of Other Detection Methods.** There are other techniques in out-of-distribution detection which require other assumptions such as more training data. For instance, Hendrycks et al. (2019a); Mohseni et al. (2020) use additional training data labeled as out-of-distribution, and the MaxLogit technique can be naturally extended should such data be available. Hendrycks et al. (2019c) use rotation prediction and self-supervised learning, but we found that scaling this to the ImageNet multiclass setting did not produce strong results. The MSP baseline trained with auxiliary rotation prediction has an AUROC of 59.1%, and with MaxLogit it attains a 73.6% AUROC, over a 10% absolute improvement with MaxLogit. Nonetheless this technique did not straightforwardly scale, as the network is better without auxiliary rotation prediction. Likewise, Lee et al. (2018b) propose to use Mahalanobis distances, but in scaling this to 1000 classes, we consistently encountered NaN errors due to high condition numbers. This shows the importance of ensuring that out-of-distribution techniques can scale.

ODIN Liang et al. (2018) assumes that, for each OOD example source, we can tune hyperparameters for detection. For this reason we do not evaluate with ODIN in the rest of the paper. However, for thoroughness, we evaluate it here. ODIN uses temperature scaling and adds an epsilon perturbation to the input in order to separate the softmax posteriors between in- and out-of-distribution images; we set these hyperparameters following DeVries & Taylor (2018). Then, MaxLogit combined with ODIN results in an FPR95 of 33.6, an AUROC of 88.8 and an AUPR of 51.3 on ImageNet. On Places365, the FPR95 is 35.3, the AUROC is 86.5, and the AUPR is 24.2. Consequently, techniques built with different assumptions can integrate well with MaxLogit. We do not train ImageNet-21K models from scratch with these methods due to limited compute.

## C  Multi-Label Out-Of-Distribution Dataset List

For multi-label classification experiments, we choose the following classes from ImageNet-21K to serve as out-of-distribution data: dolphin (n02069412), deer (n02431122), bat (n02139199), rhino (n02392434), raccoon (n02508213), octopus (n01970164), giant clam (n01959492), leech (n01937909), Venus flytrap (n12782915), cherry tree (n12641413), Japanese cherry blossoms (n12649317), red wood (n12285512), sunflower (n11978713), croissant (n07691650), stick cinnamon (n07814390), cotton (n12176953), rice (n12126084), sugar cane (n12132956), bamboo (n12147226), and tumeric (n12356395). These classes were hand-chosen so that they are distinct from VOC and COCO classes.

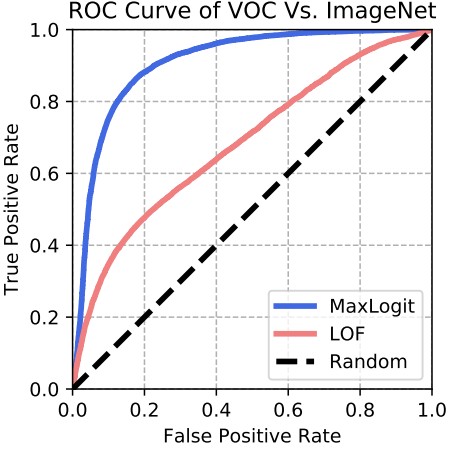

Figure 6: ROC curve with VOC as ($\mathcal{D}_{\text{in}}$) and non-overlapping ImageNet classes as ($\mathcal{D}_{\text{out}}^{\text{test}}$). Curves correspond to an uninformative "Random" detector, Local Outlier Factor, and the MaxLogit detector.

## D  OOD Segmentation

We cover methods used in the paper in more depth and the modifications necessary to make the methods work with OOD detection in semantic segmentation. We use $f$ to denote the function typically a neural network, $x$ is the input image, and $y_{i,j}$ is the prediction for pixel $i, j$. We will denote the output probability distribution per pixel as $P$ and locations $i, j$ as the location of the

respective pixel in the output. $f(x)_{i,j}$ denotes the $i$th row and $j$'th column of the output.

**Confidence Estimation.** The method proposed in DeVries & Taylor (2018) works by training a confidence branch added at the end of the neural network. We denote the network predictions as both $P$ and $\hat{c}$ whereby every pixel is assigned a confidence value.

$$b \sim B(0.5)$$
$$c := \hat{c} \cdot b + (1 - b)$$
$$P := P \cdot c + (1 - c)y$$

The confidence estimation denoted by $c$ is given "hints" during training to guide what it is learning. The $B$ is a beta distribution and acts as a regularizer similar to dropout so that the network $f$ does not exclusively rely on the true labels being present. The final loss is modified to include the extra term below:

$$\mathcal{L}_p = \frac{1}{|P|} \sum_i -\log(p_i)y_i$$
$$\mathcal{L}_c = \frac{1}{|P|} \sum_i -\log(\hat{c}_i)$$
$$\mathcal{L} = \mathcal{L}_p + \lambda \mathcal{L}_c$$

The reasoning for $\mathcal{L}_c$ is to encourage the network to output confident predictions. Finally $\lambda$ is initialized to 0.1 and is updated by a "budget" parameter which is set to the default of 0.3. The update equation:

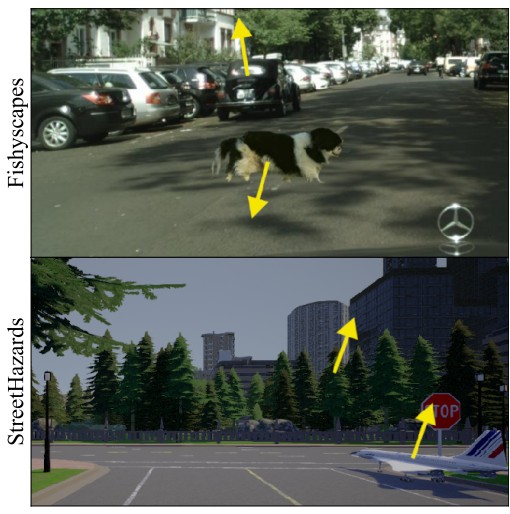

Figure 7: A comparison of lighting consistency in the Fishyscapes anomaly segmentation benchmark and our new StreetHazards dataset. The arrows point in the manually estimated direction of light on parts of the scene. In Fishyscapes, inconsistent lighting allows forensics techniques to detect the anomaly (Johnson & Farid, 2005). Unlike cut-and-paste anomalies, the anomalies in our Street-Hazards dataset are naturally integrated into their environment with proper lighting and orientation, making them more difficult to detect.

$$\begin{cases} \lambda/0.99 & \sum \hat{c}_i \leq \text{budget} \\ \lambda/1.01 & \sum \hat{c}_i > \text{budget} \end{cases}$$

This adaptively adjusts the weighting between the two losses and experimentally the update is not sensitive to the budget parameter.

**Semantic Segmentation BDD Anomalies Dataset List.** The BDD100K dataset contains 180 instances of the train class, 4296 instances of the motorcycle class, and 10229 instances of the bicycle class.

**StreetHazards 3D Models Dataset List.** For semantic segmentation experiments, we choose to use the following classes 3D models from Model Bank Library to serve as out-of-distribution data: Meta-categories: Animals, Vehicles, Weapons, Appliances, Household items (furniture, and kitchen items), Electronics, Instruments, and miscellaneous. The specific animals used are kangaroos, whales, dolphins, cows, lions, frogs, bats, insects, mongooses, scorpions, fish, camels, flamingos, apes, horses, mice, spider, dinosaurs, elephants, moose, shrimps, bats, butterflies, turtles, hippopotamuses, dogs, cats, sheep, seahorse, snail and zebra. The specific vehicles used are military trucks, motorcycles, naval ships, pirate ships, submarines, sailing ships, trolleys, trains, airplanes, helicopters, jets, zeppelin, radar tower, construction vehicles (loaders, dump trucks, bulldozer), farming vehicles (harvester, gantry crane, tractor), fire truck, tank, combat vehicles, and trailers. The specific weapons used are guns, missiles, rocket launchers, and grenades. The appliances used are refrigerators, stoves,

washing machines, and ovens. The household items used are cabinets, armoire, grandfather clocks, bathtubs, bureaus, night stand, table, bed, bookcase, office desk, glasses (drinking), throne chair, kitchen utensils (knives, forks, spoons), sofa, clothing iron, plates, sewing machine, and dressing mirror. The electronics used are computer monitor, computer mouse, hair dryer, The instruments category includes bassoon, clarinet, drums, guitar, violin, harp, and keyboard. The miscellaneous category includes rocket, space capsule, space shuttle, lunar module, glasses (wearable), weight machine, balance beam, bench press, bowling ball and pins, and pens. Several categories and instances were excluded from Model Bank Library due to their occurrence in the simulation environment such as playground equipment and various types of foliage and trees. The sizes of instances used in the dataset might not reflect the actual scale that would otherwise naturally occur. Similarly the location of instances in the dataset are not necessarily reflective of where they are likely to occur in nature.

