# OpenReview forum: "Improving and Assessing Anomaly Detectors for Large-Scale Settings"
_ICLR.cc/2022/Conference — ICLR 2022 Submitted_

### Official Review · Reviewer_Tkux · 2021-10-19

**Correctness:** 3
**Technical Novelty And Significance:** 2
**Empirical Novelty And Significance:** 4
**Recommendation:** 5
**Confidence:** 3

**Main Review:**

Strengths:
1. This paper is well motivated, which aims to push out-of-distribution (ODD) detection from small-scale settings to large-scale and real-world settings.
2. This works seems solid, which explores three large-scale settings and introduces two new datasets.
3. The proposed method (MaxLogit) shows good performance in all settings.

Weaknesses:
1. This manuscript is not very well written, some contents of this manuscript are not clearly organized. It would be better to list contributions for three different settings, and introduce datasets and methods/findings more separately.
2. The proposed method seems simple, but it's still not very clear how it works in three settings. More specific descriptions of the method would be better.

**Summary Of The Paper:**

This work explores out-of-distribution (ODD) detection in three large-scale settings: multi-class OOD detection, multi-label OOD detection and anomaly segmentation. To facilitate large-scale experiments, it introduces a novel species dataset and a road anomaly dataset for multi-class OOD detection and anomaly segmentation respectively. It also demonstrates a new setup for multi-label OOD detection. In addition, this work establishes a new baseline via a simple detector based on the maximum logit in all the three large-scale settings.

**Summary Of The Review:**

This paper is well motivated and solid, but the writing needs to be improved.

---

> ### Author Response · Authors · 2021-11-23
> **Response to Reviewer Tkux**
>
> Thank you for your careful analysis of our work. We hope the following response addresses your concerns.
>
> **Paper Organization**
>
> > “It would be better to list contributions for three different settings, and introduce datasets and methods/findings more separately.”
>
> The paper is currently organized such that the three different settings (multiclass, multi-label, and anomaly segmentation) each receive their own section. Each section is largely self-contained, independently introducing the relevant datasets, methods, experimental setup, and results.
>
> **Contribution of MaxLogit Detector**
>
> The MaxLogit detector works consistently across many problem settings and does not require substantial engineering. In practice, methods that are reliable, simple, and effective are highly preferred over complicated methods that require significant tuning or even training new models. Hence, the simplicity of our method is not a weakness but a strength.
>
> The utility of the maximum logit could not be appreciated in the small-scale settings of prior work. For example, using the small-scale CIFAR-10 setup of Outlier Exposure, the MSP attains an average AUROC of 90.08% while the maximum logit attains 90.22%, a minor 0.14% difference. However, in a large-scale setting, the difference can be *over 10%* on individual OOD datasets. We have added this analysis to the updated paper. Note that we are not claiming that the maximum logit is a mathematically sophisticated formulation, only that it serves as a consistently powerful baseline for large-scale settings that went unappreciated in small-scale settings.
>
> **Expanded Definition of MaxLogit**
>
> In the updated paper, we expand the definition of the MaxLogit detector in Section 3 into a formal definition. Section 4 provides a formal definition in the multi-label setting. For anomaly segmentation, we apply OOD detection methods to each pixel as mentioned in Section 5.1. Thank you for your suggestion. If we addressed the thrust of your concerns, we kindly ask that you consider raising your score.

---

### Official Review · Reviewer_osoG · 2021-10-31

**Correctness:** 4
**Technical Novelty And Significance:** 3
**Empirical Novelty And Significance:** 3
**Recommendation:** 5
**Confidence:** 4

**Main Review:**

Paper strengths:

+ The paper spots the reliability issue of the maximum softmax probability (MSP) on large-scale datasets.

+ The paper addresses a limited explored problem, large-scale OOD.

+ The paper is well-written and easy to follow.

+ The proposed anomaly MaxLogit score (negative of the maximum unnormalized logit for an anomaly score) leads to good performance for large-scale OOD.


Paper weaknesses:

- The MaxLogit score is nowhere formally defined. It's straightforward how is computed but a formal distribution is necessary since it composes a major contribution to the paper.

- It's not clear whether the proposed setup is near- or far-ODD setup. This claim could be well-supported by a corresponding analysis.

- The paper focuses only on the maximum softmax probability metric for the evaluations. However, there are other methodologies such as ODIN (Enhancing The Reliability of Out-of-distribution Image Detection in Neural Networks, 2018), Mahalanobis distance (A Simple Unified Framework for Detecting Out-of-Distribution Samples and Adversarial Attacks, 2018) or OECC (Outlier Exposure with Confidence Control for Out-of-Distribution Detection, 2021). It is important to include one more baseline for comparisons.

- Multi-label OOD: There is already the setup with ImageNet-1k as out-of-distribution (A benchmark for anomaly segmentation, 2019). It would be more helpful to follow an existing protocol next to the proposed one.

- Small-scale evaluation: it would be useful to see how the proposed metric performs on small-scale evaluations. There is not a single standard evaluation on this setup although approaches from the small-scale evaluation are used, e.g. MSP.


Improvements:

- "In contrast to medical anomaly segmentation and fault detection, we consider complex images from street scenes." -> This argument could be reformulated since medical imaging is quite challenging.

**Summary Of The Paper:**

The paper presents the negative of the maximum unnormalized logit (MaxLogit) as an anomaly score for out-of-distribution (OOD) detection. Also, it introduces a large-scale setup for ODD. The proposed metric shows promising results compared to the maximum softmax probability (MSP) in the proposed setup (in-distribution ImageNet-1K and out-distribution Places365). For the multi-label experiment, the PASCAL VOC and MS-COCO are in-distribution and ImageNet-22K out-distribution. The proposed MaxLogit works better than in this MSP too. Finally, the proposed metric shows promising results in the  CAOS benchmark.

**Summary Of The Review:**

Overall the paper proposes a metric and a large scale setup of out-of-distribution detection. Both contributions are valuable and can be considered novel too. There are few a points to be addressed in the paper structure, as discussed above. The major issue is to add more related approaches in the comparisons. Moreover, the small-scale evaluation would be useful to show the generalization of the proposed metric to different types of setups. Even if MaxLogit does not deliver state-of-the-art results on a small scale, it would still be important to show the results and interpret the outcome.


Post-rebuttal:

The rebuttal showed that there are still changes to be performed in the paper. It would need further work before acceptance.

---

> ### Author Response · Authors · 2021-11-23
> **Response to Reviewer osoG**
>
> Thank you for your careful analysis of our work. We hope the following response addresses your concerns.
>
> **Expanded Definition of the MaxLogit Detector**
>
> We have provided an expanded formal definition of the MaxLogit detector in Section 3 of the updated paper. Note that Section 4 already provides a formal definition in the multi-label case. Thank you for alerting us to the fact that it was not formally defined in Section 3.
>
> **We Evaluate Many Detection Methods**
>
> While the strongest baseline is the MSP, we compare to a number of other detection methods as well, including the confidence branch detector of [1], dropout epistemic uncertainty [2], and ODIN in the supplementary material. We will add comparisons to more methods for a camera-ready paper. Currently, we compare to 12 distinct OOD detection methods.
>
> **Multi-Label OOD Experimental Setup**
>
> We are glad that you identified the contribution of “A Benchmark for Anomaly Segmentation,” an incomplete draft with different contributors that misses several important components, including the Species dataset, experiments with Vision Transformers, and additional anomaly segmentation experiments. Note that this earlier draft has proven valuable for numerous followup works [3, 4] who have used the datasets and built on the MaxLogit detector. We will include discussion of these works in a camera-ready paper.
>
> **Small-Scale Evaluation of MaxLogit**
>
> The utility of the maximum logit could not be appreciated in the small-scale settings of prior work. Using the small-scale CIFAR-10 setup of [5], the MSP attains an average AUROC of 90.08% while the maximum logit attains 90.22%, a minor 0.14% difference. However, in a large-scale setting, the difference can be over 10% on individual OOD datasets. Hence, the benefit that MaxLogit provides was previously hidden. We have added this analysis to the updated paper.
>
> **Additional Changes**
>
> > "In contrast to medical anomaly segmentation and fault detection, we consider complex images from street scenes." -> This argument could be reformulated since medical imaging is quite challenging.
>
> We agree that medical anomaly segmentation is challenging, and we will change this argument. Thank you for bringing this to our attention. If we addressed the thrust of your concerns, we kindly ask that you consider raising your score.
>
>
> [1]: “Learning confidence for out-of-distribution detection in neural networks”. Devries and Taylor. arXiv 2018.
>
> [2]: “Bayesian SegNet: Model Uncertainty in Deep Convolutional Encoder-Decoder Architectures for Scene Understanding”. Kendall et al. BMCV 2017.
>
> [3]: “Standardized Max Logits: A Simple yet Effective Approach for Identifying Unexpected Road Obstacles in Urban-Scene Segmentation”. Jung et al. ICCV 2021.
>
> [4]: “Synthesize then Compare: Detecting Failures and Anomalies for Semantic Segmentation”. Xia et al. ECCV 2020 (Oral).
>
> [5]: “Deep Anomaly Detection with Outlier Exposure”. Hendrycks et al. ICLR 2019

---

> ### Author Response · Authors · 2021-12-01
> **Additional Response to Reviewer osoG**
>
> Thank you for considering our rebuttal. Here are a few remaining points that we hope are informative.
>
> **Near or Far OOD**
>
> We introduce a total of 4 datasets to help foster new work in anomaly detection for large-scale settings. As large-scale settings are already challenging, we do not specifically design our datasets to be near-distribution challenges. For example, in the multiclass experiments, we design our new Species dataset to be disjoint with ImageNet-22K and find that current anomaly detectors struggle in this setting. As Figure 2 shows, the examples in Species are quite visually distinctive and unlike images that are typical of ImageNet. Hence, it is not a near-distribution setting, but it remains challenging for current methods.
>
> **Changes to the Paper**
>
> While we aim to add some comparisons to additional methods for a camera-ready, please note that *the changes to the paper are not substantial*. As we mention in our initial response to your review, most of the changes that we have already made and plan to make are for minor clarifications, such as expanding the definition of the MaxLogit detector in Section 3 and adding a discussion of work that has already found our datasets useful and built on the MaxLogit detector. The only additional change, which reviewer qdA5 helped us identify, is to remove a small set of erroneously labeled or ambiguous examples from BDD100K, which we use to build BDD-Anomaly. This only affects 3.9% of the BDD-Anomaly examples, so is a minor modification. We hope you consider these factors in your final decision. Thank you again for your careful analysis of our work.

---

### Official Review · Reviewer_9GvF · 2021-11-02

**Correctness:** 2
**Technical Novelty And Significance:** 1
**Empirical Novelty And Significance:** 2
**Recommendation:** 3
**Confidence:** 5

**Details Of Ethics Concerns:**

No ethics concerns are found

**Main Review:**

Strengths.
+ Three three large-scale OOD settings are presented, together with datasets for empirical evaluation
+ A set of experiments is given on all three settings to provide baseline for future work
+ A anomaly scoring method called MaxLogit, which is a minor variant of the popular method MSP, is introduced, and shows effective performance on the three settings

Weaknesses.
+ Several key claims of the paper are incorrect. For example, rather than what it is claimed in the paper about small-scale multi-class OOD detection, there have been increasing efforts put on large-scale multi-class settings, such as [1-2], but the paper fully ignores those closely related work; in terms of large-scale anomaly segmentation settings, the datasets and tasks introduced in [3] clearly have more advantageous features (dataset scale, diversity of anomalies, realistic of the settings, etc.) than the proposed one in this work.
+ All the proposed datasets are created by some simple combination of existing benchmarks, wherein I cannot find any major contributions
+ The presented method MaxLogit is a trivial variant of the popular method MSP, so no major technical novelties are presented
+ The competing methods across all three settings are outdated models or simple baselines. Without comparison to recent, state-of-the-art models, it is difficult to evaluate how the presented method advances the area.

References\
[1]  "Are out-of-distribution detection methods effective on large-scale datasets?." arXiv preprint arXiv:1910.14034 (2019).\
[2] "MOS: Towards Scaling Out-of-distribution Detection for Large Semantic Space." In Proceedings of the IEEE/CVF Conference on Computer Vision and Pattern Recognition, pp. 8710-8719. 2021.\
[3] "SegmentMeIfYouCan: A Benchmark for Anomaly Segmentation." arXiv preprint arXiv:2104.14812 (2021).

**Summary Of The Paper:**

The work designs three large-scale settings for evaluating out-of-distribution detection and performs empirical evaluation on the settings to establish some baselines for future work.

**Summary Of The Review:**

Overall, the paper presents some good empirical results of large-scale OOD detection in three settings, but the work ignores a number of closely related studies and it is weak in terms of both technical novelty and empirical evaluation, thus not good enough for publication at ICLR.

---

> ### Author Response · Authors · 2021-11-23
> **Response to Reviewer 9GvF**
>
> Thank you for your careful analysis of our work. We hope the following response addresses your concerns.
>
> **Clarifying the Contribution of the MaxLogit Detector**
>
> The utility of the maximum logit could not be appreciated in the small-scale settings of prior work. For example, using the small-scale CIFAR-10 setup of [1], the MSP attains an average AUROC of 90.08% while the maximum logit attains 90.22%, a minor 0.14% difference. However, in a large-scale setting, the difference can be *over 10%* on individual OOD datasets. We have added this analysis to the updated paper. Note that we are not claiming that the maximum logit is a mathematically sophisticated formulation, only that it serves as a consistently powerful baseline for large-scale settings that went unappreciated in small-scale settings.
>
> Beyond strong empirical results, the improvements from the maximum logit are also interpretable. In Figure 3 and Section 5.1, we provide intuition for why MaxLogit performs so well in large-scale settings. Namely, when multiple classes are vying for the top prediction (e.g. with fine-grained distinctions or on class boundaries), softmax normalization can reduce the MSP as multiple classes while the maximum logit remains high. Note that this issue did not appear in previous, small-scale settings, and we identify the maximum logit as a simple but effective baseline to improve performance in large-scale settings.
>
> **Evaluating More Baselines**
>
> In the current paper, we focus on commonly used image-level OOD detection baselines, which provide a basis of comparison for future work with stronger methods. In a camera-ready paper, we plan to include comparisons to newer baselines, including those that build on the MaxLogit detector and were developed using our StreetHazards dataset [1, 2].  In total we evaluate 12 distinct baselines.
>
>
> **Creating Our Datasets Required Substantial Effort**
>
> We agree that the multi-label dataset in Section 4 is a straightforward use of existing data, and hence we do not claim any significant contribution for this in the paper. For the other datasets we introduce, substantial effort was required.
>
> For StreetHazards, we had to interface with the Unreal Engine. Working with simulators is a very time-consuming task that requires substantial engineering expertise. It took 6 months to extend the Unreal Engine at the C++ level to allow inserting anomalous objects in a realistic manner and prevent the engine from crashing.
>
> For Species, we manually curated hundreds of classes from the iNaturalist website such that they have no overlap with the extensive label space of ImageNet-21K. We also had to clean the Species dataset, which originally contained approximately 1 million images, to remove images of low quality (e.g. images with lots of foliage that occludes the anomalous animal). We will add additional details about the effort involved in curating our datasets in a camera-ready version of the paper.
>
>
> **Additional Related Work**
>
> We recognize that other work has explored anomaly segmentation and cite many such papers. Compared to datasets presented in recent works such as RoadAnomaly21 [3], the CAOS benchmark is in fact far larger with a much more diverse set of anomalies. Note that we already include results on the original RoadAnomaly dataset (containing 60 images) in Figure 5. RoadAnomaly21 increases the number of images to 100 with 26 different anomalies. By comparison, StreetHazards has more than an order of magnitude more images and anomalous object types (1500 test images and 250 anomaly types). We will include comparisons to this dataset and others (e.g. WildDash 2) in the updated paper.
>
> Relatively fewer works have explored large-scale OOD detection in the multiclass and multi-label settings. Recent published work using representations pre-trained on ImageNet-22K makes methodological errors, which we correct with the Species dataset [4], enabling future research in this important setting. We will add citations for [5, 6]. Thank you for bringing these to our attention. If we addressed the thrust of your concerns, we kindly ask that you consider raising your score.
>
>
> [1]: “Standardized Max Logits: A Simple yet Effective Approach for Identifying Unexpected Road Obstacles in Urban-Scene Segmentation”. Jung et al. ICCV 2021.
>
> [2]: “Synthesize then Compare: Detecting Failures and Anomalies for Semantic Segmentation”. Xia et al. ECCV 2020 (Oral).
>
> [3] "SegmentMeIfYouCan: A Benchmark for Anomaly Segmentation." arXiv preprint arXiv:2104.14812 (2021).
>
> [4]: “Exploring the Limits of Out-of-Distribution Detection”. Fort et al. NeurIPS 2021
>
> [5] "Are out-of-distribution detection methods effective on large-scale datasets?." arXiv preprint arXiv:1910.14034 (2019).
>
> [6] "MOS: Towards Scaling Out-of-distribution Detection for Large Semantic Space." In Proceedings of the IEEE/CVF Conference on Computer Vision

---

### Official Review · Reviewer_qdA5 · 2021-11-02

**Correctness:** 3
**Technical Novelty And Significance:** 2
**Empirical Novelty And Significance:** 3
**Recommendation:** 5
**Confidence:** 4

**Main Review:**

Strengths

S1. Species may be a valuable tool for evaluating OOD detection approaches

S2. StreetHazards is recognized as a valuable tool for evaluating dense outlier detection in several pieces of previous work.

Weaknesses

W1. max-logit has not been compared with its differentiable counterpart (log-sum-exp logit). Previous work shows that logsumexp logit can be interpreted as likelihood of the input [gratwohl00iclr] and that it can be used as a principled criterion for outlier detection [liu20nips]. A comparison with standardized max-logit [jung21iccv] would increase the value of the manuscript.

W2. experimental evaluation leaves to desire:
- Tables 1 and 2 do not contain recent baselines such as ODIN, generalized ODIN, logsumexp logit, standardized max-logit
- Table 3: it is unclear why do max-logit and max-sigmoid-logit deliver such different rankings
- Table 4 does not contain recent baselines, cf. [grcic21visapp] and the references within
- Figure 5 should report AP and FPR95 (the official benchmark metrics) in order to allow comparison across the leaderboard; it would be a good idea to include these metrics into all other tables since they are also used in Segment Me If You Can.

W3. BDD anomaly has a poor choice of the OOD class since parts of trains can occur on buses and parts of motorcycles can occur on cars. Thus many OOD pixels get very low OOD scores. Such occurences devastate the AP score and cause all approaches to come out equally bad. Thus, Fishyscapes, Segment Me If You Can and Wilddash 2 present better opportunities for dense OOD detection

Suggestions
- some details are missing or unclear: which Pascal VOC (2007 or 2012?), why does the training use a small batch size so that batchnorm has to be freezed, which batch size has been used, why does Table 2 show per-category results.
- regarding "the object’s exact class is difficult to determine": this problem also arises at semantic borders when performing dense OOD detection [grcic21visapp]
- it would be a good idea to describe WildDash 2

[jung21iccv] https://arxiv.org/abs/2107.11264

[gratwohl00iclr] https://arxiv.org/pdf/1912.03263.pdf

[liu20nips] https://arxiv.org/pdf/2010.03759.pdf

[grcic21visapp] https://arxiv.org/pdf/2011.11094.pdf

**Summary Of The Paper:**

The paper presents a collection of somewhat disjoint contributions to outlier detection. First, the authors propose Species - a novel OOD test dataset. The main advantage of this dataset is being disjoint from ImageNet-22k. Second, the authors propose to detect outliers according to the max-logit criterion. The authors claim that max-logit is especially suitable for OOD detection in multi-label environments. Third, the authors propose two novel datasets for dense outlier detection. StreetHazards is especially interesting since it allows proper rendering of introduced outliers.

**Summary Of The Review:**

Creating datasets requires a lot of hard work. Yet, the weaknesses outweigh advantages. Hence, I must conclude that this manuscript is below ICLR acceptance threshold.

---

> ### Author Response · Authors · 2021-11-23
> **Response to Reviewer qdA5**
>
> Thank you for your careful analysis of our work. We hope the following response addresses your concerns.
>
> **Evaluating More Baselines**
>
> For multiclass OOD detection, we compare to ODIN in the supplementary material. However, ODIN assumes that for each OOD example source, we can tune hyperparameters for detection. For this reason we do not evaluate with ODIN in the rest of the paper.
>
> For anomaly segmentation, we focus on strong baselines for image-level anomaly detection, providing a basis of comparison for future work. We will add more anomaly segmentation baselines for a camera-ready, including results from [1] and [2]. Note that these works already evaluate on StreetHazards, building on the strong MaxLogit baseline presented in our work.
>
> We agree that the similarity between the maximum logit and log-sum-exp energy score is interesting. Note that MaxLogit was developed as a practical solution to problems we encountered when scaling up OOD detection to large-scale settings (e.g. see Figure 3), which is entirely independent of the EBM intuition behind the log-sum-exp score. We will include a comparison to this baseline for a camera-ready thanks to your suggestion.
>
> **Experimental Evaluation Details**
>
> > “Table 3: it is unclear why do max-logit and max-sigmoid-logit deliver such different rankings”
>
> We are not sure which methods you are referring to as having different rankings. The ranking between MSP and MaxLogit is the same in Tables 2 and 3 for all metrics.
>
> In Figure 5, the AP and FPR95 are as follows:
>
> FS Lost and Found
>
> MSP        | AP: 6.0,  FPR95: 45.6
>
> MaxLogit | AP: 18.8, FPR95: 38.1
>
> Road Anomaly
>
> MSP        | AP: 20.6, FPR95: 68.4
>
> MaxLogit | AP: 24.4, FPR95: 64.9
>
> MaxLogit outperforms MSP across the board. We will add these results to the paper.
>
> **We Consider Anomalous Objects, Not Patches**
>
> We agree that certain object parts are shared between OOD objects and in-distribution objects in BDD-Anomaly (e.g. motorcycles and cars both have wheels). However, this is not problematic, because our task is to detect entire anomalous objects, not anomalous patches, and is hence well-defined. Note that standard semantic segmentation also contends with the fact that motorcycles and cars both have wheels, yet is able to distinguish the objects as a whole.
>
> **Additional Details**
>
> For multi-label experiments, we use PASCAL VOC 2012 and freeze batch norm due to GPU memory constraints. We will add these experimental details to the paper.
>
> The per-category results for Species convey additional information that may be interesting to readers. For instance, if a method were to perform substantially better than baselines on only one category then that might indicate data leakage. MaxLogit substantially outperforms MSP on all categories of Species, indicating a real improvement.
>
> We will add a discussion of WildDash 2 to the related work. Thank you for suggesting this. If we addressed the thrust of your concerns, we kindly ask that you consider raising your score.
>
>
> [1] Dense open-set recognition with synthetic outliers generated by Real NVP. Grcić et al. VISAPP 2021.
>
> [2] Standardized Max Logits: A Simple yet Effective Approach for Identifying Unexpected Road Obstacles in Urban-Scene Segmentation. Jung et al. ICCV 2021.

---

> > ### Comment · Reviewer_qdA5 · 2021-11-26
> > **On Table 3**
> >
> > The misunderstanding around Table 3 has arisen due to ambiguity of the term ranking. The review referred to the ranking of scores (eg. max-logit) used for determining the AP metric. The authors assumed the review referred to the ranking of methods. The review should have been more clear.
> >
> > It appears that the confusion has been caused by ambiguous descriptions in the paper. The caption of Table 3 states that MSP stands for "maximum softmax probability". However, the text states "MSP denotes a natural extension of the maximum softmax probability detector in the multi-label setting, obtained by taking the sigmoid of each output score f(x)_i and computing −max_i σ(f (x)_i)." It appears that the text and the caption contradict each other.
> >
> > It is not easy to guess which of these options is the correct one. If MSP is max-sigmoid-logit (as written in the text and assumed by the review), then it appears that max-sigmoid-logit and max-logit should give rise to the same AP since the sigmoid is a strictly increasing function. If MSP is max-sofmax-probability (as stated in the caption), then the column MSP appears meaningless since it makes no sense to apply softmax to logits trained with the multi-label loss.

---

> > > ### Author Response · Authors · 2021-11-30
> > > **Response to Reviewer qdA5**
> > >
> > > Thank you for clarifying the review. There was a miscommunication during writing, and the text should read that the multi-label MSP is a naive application of the original MSP to the multi-label setting (-max_i softmax(f)_i). As you note, this is not a good fit for the multi-label setting. The MaxLogit detector is a far better fit for the multi-label setting, and our results show that MaxLogit obtains superior results.

---

> > ### Comment · Reviewer_qdA5 · 2021-11-26
> > **On BDD anomaly**
> >
> > I still think that the choice of anomalies in the proposed BDD anomaly dataset is inappropriate. The argument about objects as a whole is not completely convincing since semantic segmentation is not aware of instances. Moreover, our models need to be able to recognize occluded and cropped object parts. I assume that many trains are only partially visible (eg. 0bcc752f-fd302777.jpg). Even a human would incur many strong false negatives in such images. This may be detrimental for the AP score and strongly undermine the suitability for evaluation of different approaches. Other benchmarks (eg. SegmentMeIfYouCan) do not have such noise.
> >
> > The argument about standard semantic segmentation being able to discriminate due to context is also not convincing. The standard setup has supervision for both classes. On the other hand, outlier detection has supervision only for inliers. It is not clear whether a model should be blamed for recognizing a train as a kind of a bus since no train was ever seen during training. Where is the border between desired generalization and undesired optimism in outliers?
> >
> > It would be different if our models had some common knowledge. I expect that future vision models will combine image data with textual sources. Such models could know that buses have tires and hence become suspicious about rail wheels. However, our current models do not have such knowledge and hence BDD anomaly appears ahead of its time.
> >
> > Finally, it would be a good idea if BDD corrected its many annotation mistakes. For instance, in image 076399f2-4c68835b.jpg, all pixels of a partially occluded van are labeled as class train. In image 0b79446c-48c74eea.jpg, all pixels of two large trees are labeled as class train. Such mistakes introduce additional noise to the AP score. The name of the corrected dataset should be changed in order to make it clear which version of the dataset is used in experiments.

---

> > > ### Author Response · Authors · 2021-11-30
> > > **Response to Reviewer qdA5**
> > >
> > > **BDD-Anomaly Annotations Are High-Quality**
> > >
> > > Please see our response to the above comment. In short, a small percentage of anomalous objects in BDD-Anomaly are ambiguously or erroneously labeled, which is due to noise in the original BDD100K dataset. We will remove these erroneous BDD100K examples, which currently make up only 3.9% of our validation set and do not substantially affect results.
> > >
> > > **Lower-Variance Evaluations Than Recent Datasets**
> > >
> > > The BDD-Anomaly dataset may enable lower variance evaluations than recent comparable datasets, owing to its large size. For example, RoadAnomaly21 from SegmentMeIfYouCan only contains 100 images, which may lead to a noisy evaluation. By contrast, BDD-Anomaly has over 800 validation images. Paired with the diversity of anomalies in our StreetHazards dataset, the proposed CAOS benchmark enables high-quality evaluation of anomaly segmentation methods.
> > >
> > > **Occluded Anomalies**
> > >
> > > > semantic segmentation is not aware of instances
> > >
> > > In typical semantic segmentation tasks, pixels are labeled as belonging to an object category if they could reasonably belong to an entire instance of the object. For example, isolated car parts on the road would typically not be labeled "car", but a bumper at the edge of the screen would be labeled "car", since it probably belongs to a car that is off screen. In a similar fashion, we consider occluded objects to be anomalous if they clearly do not belong to any of the in-distribution object categories. Thus, if a passenger train is only partly visible, then a human can clearly identify it as "not bus" and label it as anomalous.
> > >
> > > We agree that an interesting corner case is where the very front of a passenger train is in view at the edge of the screen, which may look like some types of bus. In this case, the anomaly is near-distribution and more difficult to distinguish, but the task is still well-defined.
> > >
> > > **OOD vs Long-Tail**
> > >
> > > > Where is the border between desired generalization and undesired optimism in outliers?
> > >
> > > All anomaly detection benchmarks must contend with the boundary between long-tail instances of categories and out-of-distribution examples. This is generally not an issue as long as the categories do not actually overlap, i.e. as long as there is some reasonable decision boundary that separates in-distribution from out-of-distribution. In the case of trains vs buses, there is a clear distinction, so the task is well-defined even if it may be difficult.
> > >
> > > > models could know that buses have tires and hence become suspicious about rail wheels. However, our current models do not have such knowledge
> > >
> > > We agree that discriminative models that only train on a limited set of categories may have difficulty identifying trains as anomalous. However, generative models are perfectly capable of one-class learning and do in fact have the knowledge required to become suspicious about rail wheels.

---

> > > > ### Comment · Reviewer_qdA5 · 2021-11-30
> > > > **More on BDD-Anomaly**
> > > >
> > > > The SMIYC benchmark contains more than 1000 images from Lost and Found, 442 images from RoadObstacles and 110 images in RoadAnomaly. Hence, SMIYC comes out better regarding quantity. However, quantity is not decisive.
> > > >
> > > > The main advantage of SMIYC is that it includes a variety of anomalies that are critical for the high level task: animals, rocks, road cones etc. If an autonomous car does not perceive one of those - it is going to be a disaster. Consequently, SMIYC appears as a better testbed for evaluating anomaly detection than excluding an arbitrary class from a closed-set dataset.
> > > >
> > > > I am not convinced that a generative model trained on buses should assign a low likelihood to a train, especially when the train wheels are occluded (as is often the case).
> > > >
> > > > In my view, BDD-anomaly is an unnecessary distraction to this manuscript. It does not contribute much, while preventing the reader to focus on StreetHazards which is much more interesting.

---

> > > > > ### Author Response · Authors · 2021-11-30
> > > > > **Response to Reviewer qdA5**
> > > > >
> > > > > **Obstacle Detection vs General Anomaly Segmentation**
> > > > >
> > > > > We agree that detecting obstacles on the road is of crucial importance to autonomous cars, and this is one of the motivations for our CAOS benchmark. However, we are also interested in benchmarking models on the general task of anomaly segmentation. RoadObstacles21 from SegmentMeIfYouCan [1], Lost and Found from earlier work [2], and StreetHazards are better suited for evaluating models on the specific task of detecting road obstacles, although StreetHazards also has anomalous objects that are not directly on the street.
> > > > >
> > > > > Note that general anomaly segmentation and obstacle detection are different problems that may admit different solutions. For instance, stixels are appropriate for road obstacle detection [2] but do not make sense for general anomaly segmentation. Thus, our work and that of SegmentMeIfYouCan are complimentary in that we lean towards different aspects of an important problem area.
> > > > >
> > > > > **Value Added by BDD-Anomaly**
> > > > >
> > > > > As we mention in Section 5, BDD-Anomaly compliments the StreetHazards dataset by evaluating models on real images, whereas StreetHazards evaluates models with a much greater diversity of anomalous objects. Thus, using both BDD-Anomaly and StreetHazards yields a more robust evaluation of anomaly segmentation methods.
> > > > >
> > > > > **Generative Models Can Detect Anomalies**
> > > > >
> > > > > > I am not convinced that a generative model trained on buses should assign a low likelihood to a train, especially when the train wheels are occluded
> > > > >
> > > > > Trains and buses are visually distinctive, and humans annotators are able to tell the difference even when the train wheels are occluded. This suggests that there is a decision boundary that robustly separates trains and buses. It is trivial to see that a perfect class-conditional generative model yields the Bayes-optimal classifier. Moreover, if the distributions are fully separable (as is intuitively the case for trains and buses), a simple threshold on the probability of one class suffices, motivating the use of generative models as OOD detectors. Many prior works have investigated generative models for OOD detection [3, 4, 5, 6].
> > > > >
> > > > >
> > > > >
> > > > > [1]: "SegmentMeIfYouCan: A Benchmark for Anomaly Segmentation". Chan, Lis, Uhlemeyer, and Blum et al. Thirty-fifth Conference on Neural Information Processing Systems Datasets and Benchmarks Track (Round 2). 2021
> > > > >
> > > > > [2]: "Lost and Found: Detecting Small Road Hazards for Self-Driving Vehicles". Pingerra and Ramos et al. 2016 IEEE/RSJ International Conference on Intelligent Robots and Systems (IROS).
> > > > >
> > > > > [3]: "Your Classifier is Secretly an Energy Based Model and You Should Treat it Like One". Grathwohl et al. ICLR 2020
> > > > >
> > > > > [4]: "Implicit Generation and Generalization with Energy Based Models". Du and Mordatch. NeurIPS 2019
> > > > >
> > > > > [5]: "Energy-based Out-of-distribution Detection". Liu et al. NeurIPS 2020
> > > > >
> > > > > [6]: "Unsupervised Anomaly Detection with Generative Adversarial Networks to Guide Marker Discovery". Schlegl et al. IPMI 2017

---

> > ### Comment · Reviewer_qdA5 · 2021-11-26
> > **Other mis-labeled trains in BDD**
> >
> > We have found the following mis-annotated trains in BDD train (around 50% of all train instances!):
> >
> > ```
> > 076399f2-4c68835b.jpg - seems to be either van or a bus
> > 0b79446c-48c74eea.jpg - tree
> > 1b082daa-98b669ec.jpg - buses
> > 1f2942b6-1cec2edf.jpg - seem to be buses
> > 20c18a94-4fee0000.jpg - seems to be a bus
> > 2736fdaf-6588c969.jpg - path
> > 33ac83ac-3f0d3a59.jpg - trees
> > 353c68ef-d95e98bb.jpg - might be a bus
> > 35af3da5-14d381a0.jpg - likely a bus
> > 3662219d-176386e2.jpg - pavement
> > 5ff99bb6-d7e354c2.jpg - truck or bus
> > 73fd9d6f-ba1ccb6d.jpg - vans
> > 7b3ee12a-26590001.jpg - bus
> > 92e34598-8184fbf0.jpg - truck or bus (seems to be on the road)
> > 9a58e8e4-4b400dc1.jpg - slope
> > 9d7e33d3-36c05b13.jpg - truck
> > 9dca7bcd-8d689e1a.jpg - truck
> > a3e8ca14-4af4e43e.jpg - truck
> > ad9f6825-a15140d0.jpg - truck
> > b4b68779-1c75351a.jpg - truck
> > bbeefd23-dbf4d60e.jpg - bus or truck
> > bf261d80-f339c048.jpg - part of a van?
> > ```
> >
> > Here are mis-annotated motorcycles in BDD train:
> > ```
> > 2ed6ab76-00000000.jpg - truck
> > 3e5c9400-4bf667a9.jpg - sign
> > 524d5d48-973a88c2.jpg - rider, tree
> > 5d40c3ac-c8594c2c.jpg - car
> > 8962cc60-ed6b6df8.jpg - van
> > ```
> >
> > Here are mis-annotated bicycles in BDD train:
> > ```
> > 04b33c7e-df4fc4c8.jpg - van
> > 097650e5-51f6c8d1.jpg - hedge
> > 0977bc8b-4dc8eea6.jpg - pedestrian
> > 0fa741ce-9b6dde95.jpg - car
> > 14020ffa-b52ddff6.jpg - tree
> > 1bef82fc-769efc1c.jpg - pole
> > 1ff92f74-697a077e.jpg - cars
> > 3740cd4f-953af260.jpg - cars
> > ```
> >
> > Here is the script we used to locate these annotation errors:
> > ```
> > import numpy as np
> > import PIL.Image as pimg
> > pathlib import Path
> > import matplotlib.pyplot as plt
> > ​
> > IMG_PATH = './bdd100k/seg/images/train/'
> > LBL_PATH = './bdd100k/seg/labels/train/'
> > QUERY_ID = 16 # the id of class train
> > ​
> > image_paths = list(sorted(Path(IMG_PATH).glob('*.jpg')))
> > label_paths = list(sorted(Path(LBL_PATH).glob('*.png')))
> > query_image_paths, query_label_paths = [], []
> > for label_path, image_path in zip(label_paths, image_paths):
> >     labels = np.array(pimg.open(label_path))
> >     if (labels == QUERY_ID).any():
> >         query_label_paths.append(label_path)
> >         query_image_paths.append(image_path)
> > ​
> > for train_img, train_lbl in zip(query_image_paths, query_label_paths):
> >     print(train_img)
> >     img = np.array(pimg.open(train_img))
> >     label = np.array(pimg.open(train_lbl))
> >     plt.title(train_img)
> >     plt.subplot(2, 1, 1)
> >     plt.imshow(img)
> >     plt.subplot(2, 1, 2)
> >     img[label == QUERY_ID] *= 0
> >     img[label == QUERY_ID] += np.array([0, 255, 0], dtype=np.uint8)
> >     plt.imshow(img)
> >     plt.show()
> > ```
> >
> > It would be great if the authors of the BDD dataset could correct such errors. That does not appear as a lot of work since the polygons appear mostly fine - only the class membership has to be corrected.

---

> > > ### Author Response · Authors · 2021-11-30
> > > **Response to Reviewer qdA5 (few mislabeled examples in BDD100K)**
> > >
> > > Thank you for identifying these mislabeled training examples in the original BDD100K dataset that we source BDD-Anomaly from. We were not aware of the high noise level in the training set for some categories (e.g. "train"). By manually verifying the BDD10K training and validation sets, we have found that this affects 3.9% of the validation examples in BDD-Anomaly. Hence, it does not substantially alter results. We will update the BDD-Anomaly dataset by removing these examples, and we will rerun the evaluations. Thank you again for spotting this problem, which only affects a small percentage of examples but is still important.

---

### Official Review · Reviewer_xNrS · 2021-11-08

**Correctness:** 3
**Technical Novelty And Significance:** 3
**Empirical Novelty And Significance:** 3
**Recommendation:** 6
**Confidence:** 4

**Main Review:**

Strength:
1. The paper is well-organized and easy to understand.
2. Out-of-distribution is a critical problem, the authors establish several large-scale benchmarks on classification and segmentation to facilitate future research.
3. The proposed MaxLogit is practical and the experiment results on large-scale benchmarks demonstrate its consistent efficacy.

Weakness:
1. I like the large-scale benchmarks constructed in the paper. But the proposed solution (MaxLogit) is just a simple extension of MSP. I cannot even tell it is a new method even though it can get significantly better results.
2. One interesting conclusion of the paper is the vision transformer-based method does not make OOD detection (on large-scale benchmarks) trivial. I would like to see more analysis and reasons behind this.
3. The benchmarks are interesting, but the comparison baselines are not sufficient. Please include more SOTA OOD results like [2, 3] (see more in [1]).


[1] Yang, Jingkang, et al. "Generalized Out-of-Distribution Detection: A Survey." arXiv preprint arXiv:2110.11334 (2021).
[2] Di Biase, Giancarlo, et al. "Pixel-wise Anomaly Detection in Complex Driving Scenes." Proceedings of the IEEE/CVF Conference on Computer Vision and Pattern Recognition. 2021.
[3] Lis, Krzysztof, et al. "Detecting the unexpected via image resynthesis." Proceedings of the IEEE/CVF International Conference on Computer Vision. 2019.

**Summary Of The Paper:**

The authors extend the out-of-distribution (OOD) detection from not-seen in small-scale settings to large-scale multiclass and multi-label ones. They provide large-scale benchmarks for evaluating ODD detectors on classification as well as segmentation. Additionally, they propose a simple yet strong baseline for this practical problem.

**Summary Of The Review:**

Overall, the authors study an practical problem at scale. They establish a large scale benchmark and propose a simple yet efficient approach for OOD detection. But they need to study more SOTA baselines. I would like to give an initial positive score and keep tuning during discussion period.

---

> ### Author Response · Authors · 2021-11-23
> **Response to Reviewer xNrS**
>
> Thank you for your careful analysis of our work. We hope the following response addresses your concerns.
>
> **Contribution of MaxLogit Detector**
>
> The utility of the maximum logit could not be appreciated as easily in previous work's small-scale settings. For example, using the small-scale CIFAR-10 setup of [1], the MSP attains an average AUROC of 90.08% while the maximum logit attains 90.22%, a minor 0.14% difference. However, in a large-scale setting, the difference can be *over 10%* on individual OOD datasets. We have added this analysis to the updated paper. Note that we are not claiming that the maximum logit is a mathematically sophisticated formulation, only that it serves as a consistently powerful baseline for large-scale settings that went unappreciated in small-scale settings.
>
> Beyond strong empirical results, the improvements from the maximum logit are also interpretable. In Figure 3 and Section 5.1, we provide intuition for why MaxLogit performs so well in large-scale settings. Namely, when multiple classes are in competition for the top prediction (e.g. with fine-grained distinctions or on class boundaries), softmax normalization can reduce the MSP as multiple classes while the maximum logit remains high. Note that this issue did not appear in previous, small-scale settings, and we identify the maximum logit as a simple but effective baseline to improve performance in large-scale settings.
>
> **Vision Transformers Do Not Solve OOD Detection**
>
> As you note, the results on our new Species dataset show that Vision Transformers do not make OOD detection trivial--in fact, they are no better than ResNet. The reason for this is simple: It was never the case that Vision Transformers solved OOD detection. Although recent work [2] including work accepted at NeurIPS [3] suggests otherwise, their results are inflated due to overlapping train and test sets (see Section 2 for details). We identify this unaddressed issue in the literature and propose the Species OOD test set that has no overlap with ImageNet-21K, correcting the literature and enabling future work with large-scale models pretrained on tens of thousands of classes.
>
> **Evaluating More Baselines**
>
> We focus our anomaly segmentation evaluation on strong image-level anomaly detectors, providing a basis of comparison for future work. We will add more anomaly segmentation baselines for a camera-ready. Note that the CAOS anomaly segmentation benchmark is only one component of our work, and in total we implement and evaluate 12 distinct methods. If we addressed the thrust of your concerns, we kindly ask that you consider raising your score.
>
> [1]: “Deep Anomaly Detection with Outlier Exposure”. Hendrycks et al. ICLR 2019
>
> [2]: “OODformer: Out-Of-Distribution Detection Transformer”. Koner et al. arXiv 2021
>
> [3]: “Exploring the Limits of Out-of-Distribution Detection”. Fort et al. NeurIPS 2021

---

### Decision · Program_Chairs · 2022-01-20

**Decision:**

Reject

**Comment:**

The authors focus on large scale out-of-distribution (OOD) detection for which they propose three benchmarks with multiclass and multi-label
high-resolution images. In these settings, they find that a simple extension, using maximum logits (MaxLogit), of a common baseline  maximum softmax probability (MSP), is surprisingly competitive to prior methods.

Five knowledgeable reviewers found the idea of having these novel benchmarks potentially interesting, but highlighted some issues that needs to be taken into account before the paper can be publishable.

First, reviewers highlighted how the presentation can be better organized (more structure on and stronger overall motivation for the three different contributions) as to present the three ideas in a more cohesive way and more formal in introducing methods (e.g. the MaxLogic and MSP) as to clearly highlight the technical contributions and the differences with other models.

Second, more baselines need to be introduced and therefore experiments extended. For example, the comparison with another detector, LogSumExpLogit, a relaxation of MaxLogit already used for (small-scale) OOD in the context of generative models. Authors provided in the rebuttal some preliminary experimental results but promised more (for a camera-ready) that could not be evaluated by the reviewers.

Third, the scope of the proposed benchmarks raised some concerns by some reviewers. If not in the motivation behind the task of treating whole objects as anomalies, additional care shall be put into the provided annotations. As one reviewer highlighted a certain percentage of images are mis-annotated. While this percentage is somehow low (3.9%) and should not change the empirical conclusions drawn in the paper, it highlights that the core contributions of the paper might have been rushed.